# Recorded poor insight as a predictor of service use outcomes: cohort study of patients with first-episode psychosis in a large mental healthcare database

Neha Ramu,[1] Anna Kolliakou [iD],[1] Jyoti Sanyal,[2] Rashmi Patel,[1,2] Robert Stewart[1,2]

► Prepublication and additional material is published online only. To view please visit the journal online (http://dx.doi.org/10.1136/bmjopen-2019-028929).

¹Institute of Psychiatry, Psychology and Neuroscience, King's College London, London, UK
²South London and Maudsley NHS Foundation Trust, King's College London, London, UK

**Correspondence to**
Dr Anna Kolliakou;
anna.kolliakou@kcl.ac.uk

## ABSTRACT

**Objectives** To investigate recorded poor insight in relation to mental health and service use outcomes in a cohort with first-episode psychosis.

**Design** We developed a natural language processing algorithm to ascertain statements of poor or diminished insight and tested this in a cohort of patients with first-episode psychosis.

**Setting** The clinical record text at the South London and Maudsley National Health Service Trust in the UK was used.

**Participants** We applied the algorithm to characterise a cohort of 2026 patients with first-episode psychosis attending an early intervention service.

**Primary and secondary outcome measures** Recorded poor insight within 1 month of registration was investigated in relation to (1) incidence of psychiatric hospitalisation, (2) odds of legally enforced hospitalisation, (3) number of days spent as a mental health inpatient and (4) number of different antipsychotic agents prescribed; outcomes were measured over varying follow-up periods from 12 months to 60 months, adjusting for a range of sociodemographic and clinical covariates.

**Results** Recorded poor insight, present in 46% of the sample, was positively associated with ages 16-35, bipolar disorder and history of cannabis use and negatively associated with White ethnicity and depression. It was significantly associated with higher levels of all four outcomes over all five follow-up periods.

**Conclusions** Recorded poor insight in people with recent onset psychosis predicted subsequent legally enforced hospitalisations and higher number of hospital admissions, number of unique antipsychotics prescribed and days spent hospitalised. Improving insight might benefit patients' course of illness as well as reduce mental health service use.

## INTRODUCTION

Schizophrenia and other psychotic disorders have potentially severe impacts both on individuals and society, although their course and prognosis are variable. The concept of insight has historically been challenging to define and measure. Currently, researchers and clinicians use long and short cognitive and

### Strengths and limitations of this study

► Our study included a large sample size, followed a naturalistic method of cohort identification and follow-up and applied natural language processing, a novel text extraction method, to ascertain insight.

► Measurement of insight (as a binary fixed variable) depended on this clearly having been stated in the clinical record and cannot be assumed to be identical to assessment through interview.

► Follow-up assessments were only feasible for those cases remaining in the geographic catchment area served by the Trust.

► Causal pathways between insight and clinical outcomes cannot be determined by our analysis.

clinical assessment schedules which measure unidimensional and multidimensional aspects of insight such as awareness of illness and its consequences, attribution of symptoms, acceptance of treatment and understanding of its effects[1 2] as well as cognitive notions such as self-reflection and self-certainty.[3] Other views additionally propose that insight depends on cognitive functioning and on a patient's cultural and life experiences that cannot accurately be measured through traditional objective assessments.[4]

The awareness and appreciation by an individual of their psychopathology has long been considered a determinant of outcome. Patients with poor insight are less likely to understand their illness; hence, have been found to be less likely to adhere to treatment[5–7] and/or require more extensive treatment.[8] Many studies have concluded that poor insight is associated with stigma and worse social performance; however, some have claimed that insight is not in fact directly linked to the outcome of the illness but how it is progressing.[9] Self-reported quality of life has been found to be higher in patients with poor insight; this has been suggested as secondary

to delusional beliefs,[10 11] although good insight has been found to be associated with higher risk of depression in people with schizophrenia[12] and with suicidality.[13 14] On the other hand, poor insight in schizophrenia has been associated with higher anxiety,[15] with obsessive/compulsive symptoms,[16] and with violent behaviour in some[17] but not in all[18] studies. In mania, poor insight has been associated with elation rather than irritability or psychosis.[19] However, others have concluded that there are no associations of insight either with symptoms or progression of schizophrenia.[20]

Despite the range of studies exploring insight in psychotic disorders, we could find no direct investigations of associations with service use outcomes. In a large mental healthcare data resource, we therefore sought to develop a means of extracting descriptions of insight from the text fields of clinical records and investigated whether recorded poor insight early after a first clinical presentation with psychosis predicted increased subsequent service use.

## METHODS

### Setting and data sources

The data used in this study were obtained from the South London and Maudsley National Health Service Foundation Trust (SLaM), one of Europe's largest mental healthcare organisations which provides comprehensive services across all ages and specialties to a defined geographic catchment of around 1.2 million residents within four south London boroughs (Lambeth, Southwark, Lewisham and Croydon). SLaM has used fully electronic health records for over 10 years and its Clinical Record Interactive Search (CRIS) tool, set up in 2008,[21 22] allows researcher access to deidentified data from the full record within a robust governance framework.[23]

### Exposure of interest and data extraction

CRIS has been substantially enhanced through natural language processing algorithms applied to extract constructs of interest from text fields in the source record using information extraction/named entity recognition techniques.[22 24] For this study, Text Hunter annotation software[25] was used to create training and test corpora classifying mentions of insight in the clinical record to train a supervised machine learning algorithm to recognise this automatically across the wider sample. An initial keyword search was carried out to extract sentences containing the word 'insight', and a human annotator manually categorised these as either 'good insight' (for example, when insight was described as 'clear', 'improving', 'partial', 'good', 'insightful', 'present', 'intact' and 'aware'), 'poor insight' (eg, described as 'lacking', 'poor', 'limited', 'insightless', 'absent', 'impaired', 'lost' or words to that effect) or as not relevant (ie, unclear/lengthy descriptions, unassessed insight, insight mentioned as a future goal rather

that at the present or where the level of insight was not immediately obvious). For generating training and independent test sets the algorithm, a randomly selected 1814 relevant sentences were manually annotated from all patients on CRIS with a previous diagnosis of schizophreniform or affective disorder (International Classification of Diseases 10th Revision (ICD-10) F2x or F3x), of which 788 were classified as having good insight, 826 as having poor insight and 200 as non-relevant statements. Precision (positive predictive value) and recall (sensitivity) were used as performance metrics based on conventional practice in text extraction evaluation.[26] The algorithm generated classified 'poor insight' instances with 0.73 precision (positive predictive value) and 0.83 recall (sensitivity) against the manual gold standard.

### Participants

For the analysis, a database was used which had been previously prepared via CRIS for an analysis of psychosis outcomes associated with cannabis use.[27] In summary, this comprised all 2026 individuals with first-episode psychosis who were accepted by a SLaM early intervention (EI) service between 1 April 2006 and 31 March 2013. Criteria for accepting patients in SLaM EI services follow those outlined in the 'Standards for Early Intervention in Psychosis Services—First Edition'.[28] Outcome data were collected up to 31 March 2014. All participants were assessed for outcomes within 12 months of the date of being accepted to an early intervention service (2026 person-years). Participants with sufficient follow-up data were also assessed for outcomes within 24 months (n=1738; 3476 person-years), 36 months (n=1461; 4383 person-years), 48 months (n=1185; 4740 person-years) and 60 months (n=926; 4630 person-years). Predictor, covariate and outcome variable data were obtained via CRIS. Besides insight, the following covariates were ascertained using values recorded closest to the date of being accepted by an early intervention service: age, gender, ethnicity, marital status, employment status and type of accommodation, primary diagnosis and cannabis use. Ethnicity was recorded according to categories defined by the UK Office for National Statistics and was condensed for this analysis into four groups (white, black, Asian, other). Diagnosis was recorded using the ICD-10 classification system. The derivation of cannabis use through natural language processing and its application as a covariate have been previously described.[27] Using the natural language processing algorithm described above, recorded poor insight was ascertained from case records within 1 month either side of the date each patient was accepted to the early intervention service, and this was defined as the primary exposure.

### Outcomes

We investigated the association between poor insight and the following mental healthcare outcomes: (1) number of psychiatric hospital admission, (2) any legally

enforced (compulsory) admission under the UK Mental Health Act (MHA), (3) the number of unique antipsychotics prescribed (as a proxy measure of treatment failure) and (4) the number of days spent in psychiatric hospital over a given follow-up period. The MHA is a UK statute law which allows compulsory admission for up to 28 days ('Section 2') or up to 6 months ('Section 3'). Antipsychotics used were ascertained both from structured fields and a natural language processing algorithm.[22]

## Statistical analysis

All participants were assessed for outcomes within 12 months of the date of being accepted to an early intervention service. Those with sufficient follow-up data were then also assessed for outcomes within 24, 36, 48 and 60 months of this first acceptance date (ie, different but overlapping follow-up periods). This was an identical approach to that previous adopted for analyses in these data,[27] investigating discrete periods of follow-up time rather than using survival analysis because of the non-proportionality of hazards. The sample was first described and factors associated with poor insight investigated. Regression models were then used to evaluate unadjusted and successively adjusted associations with the four outcomes over the five different follow-up periods. Owing to overdispersion, previously described for these data,[27] we aimed to assess associations with number of hospital admissions and number of unique antipsychotic medications using multivariable negative binomial regression (zero inflation having been investigated but giving rise to no meaningful difference). However, one of the models failed to converge, and so Poisson regressions were used instead. Associations with legally enforced hospitalisation were assessed using multivariable binary logistic regression. Associations with number of inpatient days within given observation periods were investigated using multiple linear regression models. Reference groups for covariates were defined as those with the highest prevalence for each variable, and missing categories were included as predictor variables so that no patients were excluded because of missing covariate data. Stata software V.13 (StataCorp Stata Statistical Software: Release 13; StataCorp, 2011) was used.

## Patient and public involvement

We did not directly incorporate patient and public involvement (PPI) into this particular analysis, but the SLaM Biomedical Research Centre Case Register used in the study was developed with extensive PPI and is overseen by committees that include service user and general public representatives.

## RESULTS
### Patients

From the cohort of 2,026 individuals, 927 (46%) had at least one recording of poor insight within one month

either side of their registration with the early intervention service. The sample characteristics and their associations with recorded poor insight are summarised in table 1. This was more common in patients aged 26-35, in those without recorded employment and those using cannabis. It was least common in patients of White ethnicity, those without recorded relationship status and those with accommodation status recorded as 'other' or 'council tenant'. Poor insight was most commonly recorded in bipolar disorder, schizophrenia and drug-induced psychosis, and least common in depression.

**Table 1** Sample characteristics and associations with poor insight (n=2026)

| Variable | Category | Number | % poor insight | $\chi^2$(df) P value |
|---|---|---|---|---|
| Age | <16 | 19 | 10.52 | 16.05 (3) <0.001 |
| | 16–25 | 1234 | 44.08 | |
| | 26–35 | 747 | 49.66 | |
| | >35 | 26 | 38.46 | |
| Gender | Male | 1295 | 45.56 | 0.05 (1) 0.814 |
| | Female | 731 | 46.10 | |
| Ethnicity | White | 616 | 37.50 | 36.21 (3) <0.001 |
| | Asian | 126 | 51.59 | |
| | Black | 1005 | 51.64 | |
| | Other | 279 | 40.14 | |
| Relationship | Married | 153 | 56.86 | 42.34 (3) <0.001 |
| | Divorced | 63 | 30.15 | |
| | Single | 1727 | 46.73 | |
| | Not recorded | 83 | 16.87 | |
| Employment | Employed | 107 | 47.67 | 13.78 (3) <0.001 |
| | Student | 144 | 36.11 | |
| | Unemployed | 427 | 40.51 | |
| | Not recorded | 1348 | 48.30 | |
| Accommodation | Owner | 14 | 50.00 | 64.75 (3) <0.001 |
| | Private tenant | 83 | 45.78 | |
| | Council tenant | 162 | 39.50 | |
| | Supported | 19 | 57.89 | |
| | Homeless | 37 | 45.94 | |
| | Other | 450 | 30.44 | |
| | Not recorded | 1261 | 51.78 | |
| Primary diagnosis | Schizophrenia | 1097 | 49.86 | 38.43 (5) <0.001 |
| | Bipolar | 100 | 61.00 | |
| | Depression | 94 | 30.85 | |
| | Schizoaffective | 35 | 42.85 | |
| | Drug-induced psychosis | 63 | 46.03 | |
| | Other | 637 | 38.61 | |
| History of cannabis use | No | 1087 | 36.25 | 85.43 (1) <0.001 |
| | Yes | 939 | 56.76 | |

**Table 2** Association between poor insight and number of hospital admissions (negative binomial regression)

| Time period evaluated | Incidence rate ratio for the association with insight (95% CIs, p value) | | | | |
|---|---|---|---|---|---|
| | Unadjusted | Adjusted age and gender | Adjusted age, gender, ethnicity, relationship | Adjusted age, gender, ethnicity, relationship, employment, accommodation | Adjusted age, gender, ethnicity, relationship, employment, accommodation, diagnosis |
| 12 months n=2026 | 1.79 (1.50 to 2.14) <0.001 | 1.81 (1.52 to 2.16) <0.001 | 1.73 (1.45 to 2.07) <0.001 | 1.77 (1.48 to 2.13) <0.001 | 1.73 (1.44 to 2.08) <0.001 |
| 24 months n=1738 | 1.57 (1.34 to 1.84) <0.001 | 1.60 (1.36 to 1.87) <0.001 | 1.53 (1.30 to 1.80) <0.001 | 1.56 (1.33 to 1.84) <0.001 | 1.51 (1.28 to 1.78) <0.001 |
| 36 months n=1461 | 1.45 (1.23 to 1.70) <0.001 | 1.46 (1.24 to 1.72) <0.001 | 1.41 (1.20 to 1.66) <0.001 | 1.43 (1.21 to 1.69) <0.001 | 1.38 (1.27 to 1.64) <0.001 |
| 48 months n=1185 | 1.40 (1.18 to 1.66) <0.001 | 1.40 (1.18 to 1.67) <0.001 | 1.34 (1.13 to 1.60) 0.001 | 1.35 (1.13 to 1.61) 0.001 | 1.31 (1.09 to 1.56) 0.003 |
| 60 months n=926 | 1.35 (1.12 to 1.64) 0.002 | 1.35 (1.12 to 1.64) 0.002 | 1.28 (1.05 to 1.55) 0.013 | 1.28 (1.05 to 1.57) 0.014 | 1.25 (1.02 to 1.53) 0.029 |

### Unadjusted and adjusted main outcomes

Associations with service use outcomes in unadjusted and multivariable analyses are described in tables 2–5. Higher numbers of hospitalisation episodes (table 2), higher odds of legally enforced hospitalisations (table 3), higher numbers of unique antipsychotics (table 4) and higher numbers of inpatient days (table 5) were all significantly associated with poor insight as measured at 12, 24, 36, 48 and 60 months. For proportions of patients (with present or absent poor insight) and each clinical outcome at 12, 24, 36, 48 and 60 months, please see online supplementary table 1.

**Table 3** Association between insight and legally enforced hospitalisation* (logistic regression)

| Time period evaluated | OR for the association with insight (95% CIs, p value) | | | | |
|---|---|---|---|---|---|
| | Unadjusted | Adjusted age and gender | Adjusted age, gender, ethnicity, relationship | Adjusted age, gender, ethnicity, relationship, employment, accommodation | Adjusted age, gender, ethnicity, relationship, employment, accommodation, diagnosis |
| 12 months n=2026 | 3.19 (2.54 to 4.00) <0.001 | 3.22 (2.57 to 4.06) <0.001 | 3.00 (2.38 to 3.79) <0.001 | 3.15 (2.48 to 4.01) <0.001 | 3.02 (2.37 to 3.85) <0.001 |
| 24 months n=1738 | 2.56 (2.07 to 3.17) <0.001 | 2.60 (2.10 to 3.22) <0.001 | 2.43 (1.95 to 3.03) <0.001 | 2.57 (2.05 to 3.23) <0.001 | 2.44 (1.94 to 3.07) <0.001 |
| 36 months n=1461 | 2.33 (1.87 to 2.91) <0.001 | 2.36 (1.89 to 2.96) <0.001 | 2.25 (1.79 to 2.84) <0.001 | 2.37 (1.87 to 3.01) <0.001 | 2.26 (1.78 to 2.88) <0.001 |
| 48 months n=1185 | 2.25 (1.76 to 2.87) <0.001 | 2.26 (1.77 to 2.89) <0.001 | 2.14 (1.66 to 2.76) <0.001 | 2.25 (1.73 to 2.92) <0.001 | 2.15 (1.65 to 2.81) <0.001 |
| 60 months n=926 | 2.06 (1.56 to 2.71) <0.001 | 2.06 (1.56 to 2.73) <0.001 | 1.89 (1.41 to 2.52) <0.001 | 2.04 (1.50 to 2.77) <0.001 | 1.95 (1.43 to 2.66) <0.001 |

*Mental Health Act Section.

**Table 4** Association between insight and number of unique antipsychotics prescribed (Poisson regression)

| Time period evaluated | Incidence rate ratio for the association with insight (95% CIs, p value) | | | | |
|---|---|---|---|---|---|
| | Unadjusted | Adjusted age and gender | Adjusted age, gender, ethnicity, relationship | Adjusted age, gender, ethnicity, relationship, employment, accommodation | Adjusted age, gender, ethnicity, relationship, employment, accommodation, diagnosis |
| 12 months n=2026 | 1.39 (1.29 to 1.49) <0.001 | 1.39 (1.29 to 1.49) <0.001 | 1.34 (1.25 to 1.44) <0.001 | 1.35 (1.25 to 1.45) <0.001 | 1.32 (1.23 to 1.42) <0.001 |
| 24 months n=1738 | 1.40 (1.30 to 1.49) <0.001 | 1.40 (1.31 to 1.50) <0.001 | 1.35 (1.26 to 1.44) <0.001 | 1.37 (1.28 to 1.47) <0.001 | 1.34 (1.25 to 1.44) <0.001 |
| 36 months n=1461 | 1.35 (1.26 to 1.45) <0.001 | 1.35 (1.26 to 1.45) <0.001 | 1.30 (1.21 to 1.40)<0.001 | 1.32 (1.23 to 1.42) <0.001 | 1.29 (1.20 to 1.39) <0.001 |
| 48 months n=1185 | 1.30 (1.21 to 1.40) <0.001 | 1.30 (1.20 to 1.40) <0.001 | 1.25 (1.15 to 1.35) <0.001 | 1.27 (1.17 to 1.37) <0.001 | 1.23 (1.14 to 1.34) <0.001 |
| 60 months n=926 | 1.29 (1.18 to 1.40) <0.001 | 1.27 (1.17 to 1.38) <0.001 | 1.21 (1.11 to 1.31) <0.001 | 1.24 (1.13 to 1.35) <0.001 | 1.21 (1.11 to 1.32) <0.001 |

## DISCUSSION

In a large cohort of cases with first-episode psychosis drawn from a mental healthcare database, we developed an algorithm to detect recorded poor insight and investigated this as a predictor of four subsequent service use outcomes. Rate of poor insight in our cohort (46%) was in the range of that reported by studies assessing it through routine data collection methods (~50%).[29–32] Poor insight was, in summary, significantly and independently associated with higher number of hospitalisation episodes, higher odds of legally enforced hospitalisation, higher numbers of days spent as an inpatient and higher numbers of unique antipsychotic agents prescribed. Associations with these outcomes remained significant over the five follow-up time-points at 12, 24, 36, 48 and 60 months .

**Table 5** Association between insight and days spent hospitalised during the observation period (linear regression)

| Time period evaluated | B-coefficient for the association with recorded poor insight (95% CIs, p value) | | | | |
|---|---|---|---|---|---|
| | Unadjusted | Adjusted age and gender | Adjusted age, gender, ethnicity, relationship | Adjusted age, gender, ethnicity, relationship, employment, accommodation | Adjusted age, gender, ethnicity, relationship, employment, accommodation, diagnosis |
| 12 months n=2026 | 26.42 (22.09 to 30.76) <0.001 | 26.77 (22.42 to 31.12) <0.001 | 25.47 (21.05 to 29.88) <0.001 | 26.33 (21.85 to 30.82) <0.001 | 25.70 (21.17 to 30.22) <0.001 |
| 24 months n=1738 | 37.47 (29.07 to 45.87) <0.001 | 38.58 (30.17 to 47.00) <0.001 | 36.57 (28.04 to 45.10) <0.001 | 38.28 (29.62 to 46.95) <0.001) | 36.78 (28.05 to 45.50) <0.001 |
| 36 months n=1461 | 46.84 (34.24 to 59.44) <0.001 | 48.55 (35.96 to 61.13) <0.001 | 46.04 (33.34 to 58.74) <0.001 | 47.32 (34.46 to 60.18) <0.001 | 45.33 (32.43 to 58.25) <0.001 |
| 48 months n=1185 | 48.64 (31.06 to 66.22) <0.001 | 50.77 (33.23 to 68.30) <0.001 | 46.09 (28.41 to 63.77) <0.001 | 47.55 (29.67 to 65.44) <0.001 | 44.77 (26.85 to 62.68) <0.001 |
| 60 months n=926 | 52.91 (29.89 to 75.93) <0.001 | 53.23 (30.19 to 76.27) <0.001 | 47.18 (23.99 to 70.37) <0.001 | 48.40 (24.79 to 72.00) <0.001 | 44.79 (21.15 to 68.43) <0.001 |

Loss of insight has long been considered a potentially important feature of psychotic disorders, and clearly establishing a therapeutic alliance is more challenging when insight is poor, accounting for associations found with reduced treatment adherence.[7 8] On the other hand, reduced awareness of a mental disorder has been suggested to a reduced personal impact of that disorder, accounting for associations found with better self-rated quality of life[10 11] and lower risk of depression and suicidality.[12–14] It is therefore understandable that there has been some controversy over whether poor insight has prognostic relevance. Our study focused on a range of outcomes derived from mental healthcare records and, as described above, found these to be worse in people recorded as having poor insight during the first 5 years of their clinical care.

Strengths of the study include the large sample size and naturalistic nature of the cohort and follow-up. It has also demonstrated the great potential for natural language processing (NLP) applied to routine healthcare records in deriving novel information of clinical relevance. However, key limitations need to be borne in mind when interpreting the findings. Considering the measurement of insight, the performance of the NLP algorithm was judged to be satisfactory and clearly represents an important step forward in routine data collection (structured fields in case records invariably fail to record this construct thus rendering it invisible in conventional healthcare databases); furthermore, suboptimal measurement of insight would have obscured rather than exaggerated the prospective associations with the outcomes of interest. However, clearly, statements about insight have to be recorded in the first place, and there may be clinical circumstances and reasons which render these more or less likely. For example, clinicians may be biased to record insight when it is poor or noticeably absent but not when it is present. In addition, as we measured insight as a fixed binary variable, the construct cannot be assumed to be identical to an assessment of insight in a research interview, and we solely focused on recorded poor insight and did not seek to subcharacterise the sample into those with mixed good or poor statements. Additionally, the precision and recall rates still allow for a risk of false positive and false negative instances of poor insight, and further work could be employed to improve the performance metrics. In terms of follow-up, hospitalisations and other outcomes would only be ascertained for those cases which remained in the geographic catchment served by SLaM, so outmigration might have affected longer interval findings. In this analysis, as with a previous analysis of cannabis use as a risk factor in this sample,[22] we investigated associations over different time periods. Longer follow-up evaluations clearly provide a more informed picture of prognosis; however, insight cannot be assumed to be constant over time and we did not attempt to quantify these trajectories—for example, more effective treatment may result in a virtuous cycle involving improved insight and better therapeutic engagement.

Residual confounding cannot be absolutely excluded, and causal pathways also remain to be elucidated; however, these might include failure to establish initial engagement with services resulting in symptomatic deterioration and requirement for inpatient care—particularly supported by the high use of legally enforced hospitalisation. It is possible that poor insight at first presentation is associated with antipsychotic treatment failure, as suggested by the higher number of antipsychotics used, although it is difficult to draw this conclusion with certainty because of potentially complex interactions between insight and treatment effects. Poor insight might place strains on social support networks and compromise the role of protective factors, accounting for the observed associations between poor insight and indicators of social/financial disadvantage in our cohort. It might result in risk behaviours which result in worse outcomes, although we adjusted for cannabis use as one of these potential pathways, and this did not account substantially for the associations observed. Finally, it is possible that poor insight is not a risk factor itself but is a marker of a disorder which is already more severe in other respects (such as symptomatically or in terms of functional deterioration). Importantly, this study focused on the relationship between insight recorded shortly after presentation and the outcomes of interest, and we did not seek to capture changes in insight over the follow-up periods; this would be a potentially useful further line of enquiry, although dependent on the extent to which fluctuations in insight are recorded in routine mental healthcare.

## CONCLUSION

Our findings do support an important prognostic role for poor insight in people with psychotic disorders when this is mentioned early after first clinical presentation. Although economic modelling was not attempted, clearly, outcomes such as the number and duration of hospitalisation episodes, number of antipsychotics prescribed and legally enforced hospitalisations have substantial impact, and measures taken to improve insight might similarly bring important benefits at a service level as well as on individuals' course of illness.

**Correction notice** This article has been corrected since it was published online. The license type has been updated from CC BY-NC to CC BY.

**Contributors** The study was conceived by NR, RS and RP. Natural language processing applications were developed by NR, AK and JJ. Analyses were carried out by NR and AK. NR and AK led the preparation of the final report, to which all authors contributed significantly, approving the final version.

**Funding** RS, RP, JJ and AK are partly funded by the National Institute for Health Research Biomedical Research Centre and Dementia Biomedical Research Unit at South London and Maudsley NHS Foundation Trust and King's College London. RP has received support from a Medical Research Council (MRC) Health Data Research UK Fellowship (MR/S003118/1) and a Starter Grant for Clinical Lecturers (SGL015/1020) supported by the Academy of Medical Sciences, The Wellcome Trust, MRC, British Heart Foundation, Arthritis Research UK, the Royal College of Physicians and Diabetes UK.

**Competing interests** None declared.

**Patient consent for publication** Not required.

**Ethics approval** Oxford C Research Ethics Committee, reference 08/H0606/71+5.

**Provenance and peer review** Not commissioned; externally peer reviewed.

**Data sharing statement** All data relevant to the study are included in the article or uploaded as supplementary information.

**ORCID iD**
Anna Kolliakou http://orcid.org/0000-0003-1234-4129

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
