## [Reviewer comments · BMJ Open]

ARTICLE DETAILS

TITLE (PROVISIONAL)	Recorded poor insight as a predictor of service use outcomes: a cohort study of patients with first episode psychosis in a large mental healthcare database
AUTHORS	Ramu, Neha; Kolliakou, Anna; Jyoti, Jyoti; Patel, Rashmi; Stewart, Robert

VERSION 1 – REVIEW

REVIEWER	Nicholas Breitborde The Ohio State University, USA
REVIEW RETURNED	29-Jan-2019

GENERAL COMMENTS	1) In the introduction, the authors state “Self-reported quality of life has been found to be higher in association with poor insight; this has been suggested as secondary to delusional beliefs, although good insight has been found to be associated with higher risk of depression in people with schizophrenia and with suicidality.” I’m a bit unclear what the authors are trying to convey with the phrase “higher in association.” Do they mean than the association in greater in magnitude? If so, greater in magnitude to what? 2) The authors provide a description the validation of the machine learning algorithm as compared to hand-coded data on pg. 5. To support the validation of the machine learning algorithm, they present the positive predictive value (PPV) and specificity of the machine learning codes compared to hand-coded data. What is the rationale for including these two measures (i.e., PPV and specificity) and not including values for negative predictive value or sensitivity? 3) In describing the sample, it would be helpful to know the eligibility criteria for the first-episode psychosis (FEP) service. This would help to facilitate comparisons (or contrasts) with data from other FEP services that may use similar or different eligibility criteria. Additionally, are data on duration of psychotic illness available for this sample? 4) The authors reports that “All participants were assessed for outcomes within 12 months.” Does this mean that outcomes were assessed: (i) after 12 months of treatment or (ii) sometime during the first 12 months of treatment? If the latter, how was between-subject variation in the duration of treatment at time of outcome assessment addressed in the analyses? 5) Participants without outcome data at 24, 36, 48, and 60 months were not included the analyses for these time-points. What is the
---

	rationale for dropping these participants from the analyses rather than utilizing statistical procedures for missing data (e.g., multiple imputation)? 6) In the section “Unadjusted and Adjusted Main Outcomes,” the authors note that the findings between insight are significant at certain time points and not at others. What do you think may account for this temporal variation in the results? 7) Might there be some utility in examining insight throughout the follow-up period? Could variation in the course of insight over the course of treatment account for the temporal variation in the strength of the associations between outcomes and insight (see #6)? 8) Could between-subject variation in treatment type (e.g., therapy, medication, family psychoeducation, etc.) or dosage (e.g., number of therapy sessions) have affected the results? Given that the authors have access to participants' medical records, could these variables be obtained and included as covariates in the analyses?
--	---

REVIEWER	Keith Gaynor School of Psychology, University College Dublin, Ireland
REVIEW RETURNED	01-Feb-2019

GENERAL COMMENTS	Review Insight Psychosis Thank you for putting forward this fascinating article. Although, insight has been a controversial concept within psychosis, the article highlights its potential prognostic value across a large naturalistic sample and over a significant period of time. The article also highlights the research potential of natural language processing which potentially has a very wide application. For this reviewer, there a number of issues which need to be addressed within text Strengths and limitations: it should be highlighted that insight was measured as a fixed binary variable Introduction: the multi-factorial nature of the concept of insight should be discussed.
--

	 • For instance,  ○ Birchwood et al. have conceptualised insight as a continuum. They have also proposed a three factor model, differentiating: awareness of illness, need for treatment and attribution of symptom. ○ Tranulis et al. (2008) highlighted divergent findings between cognitive and clinical constructs of insight. ○ McCormack et al. (2014) discussed that (1) different dimensions of the concept of clinical insight may be unstable during a first episode psychosis; (2) there is a high level of inter-rater difference; (3) each insight scale measures a different underlying concept of insight and have a low level of inter-correlation While the desire to simplify the concept of insight is understandable within the methodology, the range of issues associated with insight, its concept and measurement need to be alluded to. Method  • What is the accuracy of the Search Engine? • Is there any information whether there is a recording bias when clinicians record insight i.e. they record it when it is poor or noticeably absent but there is no record when insight is present. • Insight is defined as a binary variable neglected the large literature above indicating its dynamic, multi-model and continuous nature. This is not discussed as a weakness • How does the rate of poor insight as measured by language analysis in this sample compare to other FEP samples assessed in traditional ways Results  • What is the rate of missing data? • A consort-type flow chart would be useful Discussion  • The conceptual and methodological issues (above) should be discussed • Limitations should include the risk of false positive data given the ppv and sensitivity. • Limitations should include a binary definition of insight. • To my mind the most novel and interesting aspect of the study is the use of natural language processing and this methodology could be discussed.
--	--

VERSION 1 – AUTHOR RESPONSE

Reviewer: 1

We would like to thank Dr Breitborde for his feedback. We have addressed all the points brought forward which has hopefully given the paper more clarity and consistency.

1) In the introduction, the authors state “Self-reported quality of life has been found to be higher in association with poor insight; this has been suggested as secondary to delusional beliefs, although good insight has been found to be associated with higher risk of depression in people with schizophrenia and with suicidality.”

I'm a bit unclear what the authors are trying to convey with the phrase “higher in association.” Do they mean than the association in greater in magnitude? If so, greater in magnitude to what?

This was a typing mistake which has now been rectified.

2) The authors provide a description the validation of the machine learning algorithm as compared to hand-coded data on pg. 5. To support the validation of the machine learning algorithm, they present the positive predictive value (PPV) and specificity of the machine learning codes compared to hand-coded data. What is the rationale for including these two measures (i.e., PPV and specificity) and not including values for negative predictive value or sensitivity?

A reference has been added to support the use of these two measures as gold standard in text extraction in the ‘Methods’ section.

3) In describing the sample, it would be helpful to know the eligibility criteria for the first-episode psychosis (FEP) service. This would help to facilitate comparisons (or contrasts) with data from other FEP services that may use similar or different eligibility criteria. Additionally, are data on duration of psychotic illness available for this sample?

Data on DUP are not available. Eligibility criteria for first-episode psychosis have now been addressed in the ‘Participants’ section.

4) The authors reports that “All participants were assessed for outcomes within 12 months.” Does this mean that outcomes were assessed: (i) after 12 months of treatment or (ii) sometime during the first 12 months of treatment? If the latter, how was between-subject variation in the duration of treatment at time of outcome assessment addressed in the analyses?

5) Participants without outcome data at 24, 36, 48, and 60 months were not included the analyses for these time-points. What is the rationale for dropping these participants from the analyses rather than utilizing statistical procedures for missing data (e.g., multiple imputation)?

Points 4 and 5 – We have revised the text in ‘Participants’ section to more clearly describe the follow-up process. Multiple imputation was not an appropriate method because the outcome data are not missing, they haven't been measured as they are occurring at a future time point.

6) In the section “Unadjusted and Adjusted Main Outcomes,” the authors note that the findings between insight are significant at certain time points and not at others. What do you think may account for this temporal variation in the results?

It should be borne in mind that sample sizes differ for the different follow-up periods (and there may have been a misunderstanding that the results reflect time points rather than follow-up periods, which we have sought to avoid through amended text). We feel that changes in coefficient values are more informative than changes in p-values in this respect and have amended text to reflect this.

7) Might there be some utility in examining insight throughout the follow-up period? Could variation in the course of insight over the course of treatment account for the temporal variation in the strength of the associations between outcomes and insight (see #6)?

We agree that variation in insight over time would be of interest, although we feel that it would require specific attention with a different design and is beyond the scope of this particular paper. We have added text towards the end of the Discussion to mention this. As described above, we believe that

alterations in significance levels are most likely to reflect changes in analysed sample sizes between the different follow-up conditions, rather than temporal variation.

8) Could between-subject variation in treatment type (e.g., therapy, medication, family psychoeducation, etc.) or dosage (e.g., number of therapy sessions) have affected the results? Given that the authors have access to participants' medical records, could these variables be obtained and included as covariates in the analyses?

This is a very important point. Our results only take the research forward for a certain distance. The variety of mediating factors is vast and could include all above-mentioned reasons as well as lifestyle factors and adverse events. It would require a new focus and possibly an altogether new piece of research to address this appropriately.

Reviewer: 2

We would like to thank Dr Gaynor for his suggestions. These have been implemented.

Thank you for this paper. its methodology in particular is novel and interesting.

FORMATTING AMENDMENTS (if any)

Required amendments will be listed here; please include these changes in your revised version:

-The in text citation for 'Supplementary Table 1' is missing on your main text of your main document file. Please amend accordingly.

- Kindly remove all your Supplementary Table in your Main Document and upload it separately under file designation "Supplementary File" in PDF Format.

Suppl Table 1 has been cited in the text and uploaded in a different file.

Reviewer: 3

We would like to thank the reviewer for the comments and suggestions. We have implemented all of them and hope that the revised paper addressed these important points accurately and sufficiently. Insight Psychosis

Thank you for putting forward this fascinating article.

Although, insight has been a controversial concept within psychosis, the article highlights its potential prognostic value across a large naturalistic sample and over a significant period of time.

The article also highlights the research potential of natural language processing which potentially has a very wide application.

For this reviewer, there are a number of issues which need to be addressed within text

Strengths and limitations: it should be highlighted that insight was measured as a fixed binary variable. This has been added to 'Strengths and Limitations'

Introduction: the multi-factorial nature of the concept of insight should be discussed.

For instance,

- o Birchwood et al. have conceptualised insight as a continuum. They have also proposed a three factor model, differentiating: awareness of illness, need for treatment and attribution of symptom.

- o Tranulis et al. (2008) highlighted divergent findings between cognitive and clinical constructs of insight.

- o McCormack et al. (2014) discussed that (1) different dimensions of the concept of clinical insight may be unstable during a first episode psychosis; (2) there is a high level of inter-rater difference; (3)

each insight scale measures a different underlying concept of insight and have a low level of inter-correlation

While the desire to simplify the concept of insight is understandable within the methodology, the range of issues associated with insight, its concept and measurement need to be alluded to.

This has been added to 'Introduction'

Method

- What is the accuracy of the Search Engine?

We didn't use a search engine – the CRIS system is an interface which allows access to a de-identified copy of the full clinical record at SLAM.

- Is there any information whether there is a recording bias when clinicians record insight i.e. they record it when it is poor or noticeably absent but there is no record when insight is present.

This point has now been addressed in 'Discussion' as suggested

- Insight is defined as a binary variable neglected the large literature above indicating its dynamic, multi-model and continuous nature. This is not discussed as a weakness

This point has been addressed as per initial suggestion about the Strengths and Weaknesses section and well as 'Discussion'

- How does the rate of poor insight as measured by language analysis in this sample compare to other FEP samples assessed in traditional ways

This has been now addressed in 'Discussion'

Results

- What is the rate of missing data?
- A consort-type flow chart would be useful

The 'Participants' section has been revised to clarify the follow-up procedure

Discussion

- The conceptual and methodological issues (above) should be discussed
- Limitations should include the risk of false positive data given the ppv and sensitivity.
- Limitations should include a binary definition of insight.
- To my mind the most novel and interesting aspect of the study is the use of natural language processing and this methodology could be discussed.

These points have now been addressed in 'Discussion'

VERSION 2 – REVIEW

REVIEWER	Nicholas Breitborde The Ohio State University, USA
REVIEW RETURNED	20-Mar-2019

GENERAL COMMENTS	The reviewers have done a commendable job in addressing all of the questions from my initial review.
--